

**ARIOS: An acidification ocean database for the Iberian Upwelling Ecosystem**
**(1976 - 2018)**
**Xosé Antonio Padin\*[1], Antón Velo[1] and Fiz F. Pérez[1]**
[1,] Instituto de Investigaciones Marinas, IIM-CSIC, 36208 Vigo, Spain.
*padin@iim.csic.es, avelo@iim.csic.es, fiz.perez@iim.csic.es*
**1. Abstract**
A data product of 17,653 discrete samples from 3,343 oceanographic stations
combining measurements of pH, alkalinity and other biogeochemical parameters off the
North-western Iberian Peninsula from June 1976 to September 2018 is presented in this
study. The oceanography cruises funded by 24 projects were primarily carried out in the
*Ría de Vigo* coastal inlet, but also in an area ranging from the Bay of Biscay to the
Portuguese coast. The robust seasonal cycles and long-term trends were only calculated
along a longitudinal section, gathering data from the coastal and oceanic zone of the
Iberian Upwelling System. The pH in the surface waters of these separated regions,
which were highly variable due to intense photosynthesis and the remineralization of
organic matter, showed an interannual acidification ranging from -0.0016 yr$^{-1}$ to -0.0032
yr$^{-1}$ that grew towards the coastline. This result is obtained despite the buffering
capacity increasing in the coastal waters further inland as shown by the increase in
alkalinity by 1.1±0.7 μmol kg$^{-1}$ yr$^{-1}$ and 2.6±1.0 μmol kg$^{-1}$ yr$^{-1}$ in the inner and outer *Ría*
*de Vigo* respectively, driven by interannual changes in the surface salinity of
0.0193±0.0056 psu yr$^{-1}$ and 0.0426±0.016 psu yr$^{-1}$ respectively. The loss of the vertical
salinity gradient in the long-term trend in the inner ria was consistent with other
significant biogeochemical changes such as a lower oxygen concentration and
fertilization of the surface waters. These findings seem to be related to a growing
footprint of sediment remineralization of organic matter in the surface layer of a more
homogeneous water column.
Data are available at: http://dx.doi.org/10.20350/digitalCSIC/12498 (Pérez et al., 2020).
**2. Introduction**
CO$_2$ emissions of anthropogenic origin (fossil fuels, land use and cement
manufacturing) into the atmosphere are the main cause behind the warming of the Earth
due to the greenhouse effect (IPCC, 2013). Given the constant exchange of gases





through the air-sea interface, the oceanic reservoir plays a key role as a sink for about
31% of anthropogenic $CO_2$ emissions (Sabine et al., 2004), controlling the partial
pressure of carbon dioxide in the atmosphere and regulating global temperatures.
The $CO_2$ uptake by the oceans produces changes in the inorganic carbon system in spite
of being partially dampened by the seawater buffering capacity. This ability of seawater
to fix anthropogenic $CO_2$ becomes more limited as more $CO_2$ is absorbed, which will
make it difficult to stabilize atmospheric $CO_2$ in the future (Orr et al., 2009). In any
case, the rapid increase of $CO_2$ in the atmosphere decreases the ocean's pH (Caldeira
and Wickett, 2003; Raven et al., 2005). This effect of $CO_2$ absorption, which is known
as ocean acidification, conditions the buffering capacity of seawater and to some extent
the exchange of $CO_2$ between the ocean and the atmosphere. The Intergovernmental
Oceanographic Commission of the United Nations identified the chemical change in
seawater brought about by ocean acidification as an indicator of a stressor on marine
ecosystems with a negative impact on socio-economic activities such as fishing and
shellfish farming. Hence, it was necessary for the oceanography community to observe
and gather data about pH and other parameters of the marine carbon system to produce
global and regional data products in order to help sustainably manage the ocean's
resources.

The threat of oceanic acidification of marine ecosystems is especially significant in
regions like coastal upwelling areas, which are more sensitive and appear to respond
faster to anthropogenic perturbations (Feely et al., 2008; Gruber et al., 2012; Lachkar,
2014; Hauri et al., 2013). These ecosystems are characteristic for their complex physical
and biogeochemical interactions and for sustaining enormous biological productivity
and productive fisheries (Pauly and Christensen, 1995; Haury et al., 2009). The
photosynthetic activity in these regions is also an important mechanism for the seawater
$CO_2$ uptake, converting most of these areas into atmospheric $CO_2$ sinks (Pérez et al.,
1999; Cobo-Viveros et al., 2013). However, the great spatial and temporal variability
has been widely spaced and intermittent over time, preventing a complete view from
being obtained of ocean acidification in the upwelling system.

The effects of ocean acidification on marine ecosystems has awoken the interest of the
international community and has stimulated global impetus for gathering high quality



time-series measurements of the marine inorganic carbon system (Hofmann et al., 2011;
Andersson and MacKenzie, 2012; McElhany and Busch, 2013; Takeshita et al., 2015;
Wahl et al., 2016) and for predicting the future evolution of the pH caused by climate
change. Researchers at the *Instituto de Investigaciones Mariñas* (IIM-CSIC) in Vigo
(Spain) have already started this task, measuring the marine inorganic carbon system
and associated parameters on the Galicia coast in the northwest of the Iberian Peninsula,
since 1976. These biogeochemical changes, including the pH variation, have been
gathered via different projects with particular objectives between 40ºN and 45ºN, 11ºW
and the Galician coast over the past 40 years. The current database, hereinafter called
ARIOS (Acidification in the rias and the Iberian continental shelf) database, holds
biogeochemical information from 3,357 oceanographic stations, giving 17,653 discrete
samples. This unique collection is a starting point for evaluating the ocean acidification
in the Iberian Upwelling System characterized by intense biogeochemical interactions
as an observation-based analysis, or for use as inputs in a coupled physical-
biogeochemical model to disentangle these interactions at the ecosystem level.

**3. Data provenance**
**3.1. Region**
The main characteristic of the Galician coastline, located in the north-west of the
Iberian Peninsula, is the *Rías Baixas*, four long coastal estuaries or rias (>2.5 km$^3$)
between 42ºN and 43ºN (Fig. 1). The water exchange between the *Rías Baixas* and open
waters is drastically affected by the coastal wind pattern as part of the Canary Current
Upwelling System (Wooster et al., 1976; Fraga 1981; Arístegui et al., 2004). Under the
predominance of northeasterly winds (Blanton et al., 1984) during spring-summer, the
surface offshore transport of surface waters leads to a rising cold, nutrient-rich, deep
water mass called the Eastern North Atlantic Central Water (ENACW) (Ríos et al.,
1992). Under these conditions, the *Rías Baixas* act as an extension of the continental
shelf (Rosón et al., 1995; Souto et al., 2003; Gilcoto et al., 2017), where upwelling
filaments extending westward export primary production from the coast into the ocean
(Álvarez-Salgado et al., 2001). In the opposite direction, the prevalence of northward
winds (Blanton et al., 1984) moves the surface waters towards the coast, where they
accumulate, sink and thus isolate the coast. This process, known as downwelling, is
typical during the autumn-winter along with other characteristics such as the warm,
salty waters from the Iberian Poleward Current (IPC) of subtropical origin (Fraga et al.,



1982; Alvarez-Salgado et al., 2006) that flows constrained to the Iberian shelf break
(Frouin et al., 1990). The run-off from local rivers also contributes to the presence of
river plumes over the shelf (Otero et al., 2008). These hydrodynamic conditions, the
meteorological forcings and the alternation of periods of upwelling and downwelling
(Àlvarez, 1999; Gago et al., 2003c; Cobo-Viveros et al., 2013) stimulate the
development of intense primary production and high rates of recycling and downward
carbon export (Alonso-Pérez and Castro, 2014). The result of this biogeochemical
variability in terms of air-sea $CO_2$ exchange is that the surface waters act as a net $CO_2$
sink that is especially intense and variable over the shelf compared to offshore or in the
inner *Rías Baixas* (Padin et al., 2010).

In addition to the short-term and seasonal variability, significant changes in the long-
term scale have been reported in this region. In addition to changes such as the
weakening and shortening of the upwelling events (Lemos and Sansó, 2006; Pérez et
al., 2010; Alvarez-Salgado et al., 2009), warming (González-Pola et al., 2005; Pérez et
al., 2010) and changes in the composition of phytoplankton (Bode et al., 2009; Pérez et
al., 2010), the acidification in the surrounding waters of the Galician coast has also been
observed at a rate of -0.0164 pH units per decade in the first 700 metres (Ríos et al.,
2001; Castro et al. 2009).

**3.2. Data sources**
The ARIOS database is a compilation of biogeochemical properties with discrete
measurements of temperature, salinity, oxygen, nutrients, alkalinity, pH and chlorophyll
that were sampled in waters off the northwest of the Iberian Peninsula from 1976 to
2018 and measured by IIM-CSIC (Table 1). This data collection is part of the research
by 24 projects and oceanographic cruises conducted in response to different aims. The
different sampling strategies built up an irregular biogeochemical database whose
particular frequency and spatial coverage is shown in Figure 2.

The contribution to the ARIOS database from the oceanographic cruises and projects
over the different decades is described below.

**Cruises in the 70s and 80s:**



The first three cruises were carried out over three periods (1976, 1981-1983 and 1983-
1984), sampling the *Ría de Vigo*. These cruises were designed to provide environmental
information (upwelling events, estuarine circulation, continental inputs, etc.) for
research into the biology of some fish species. They measured identical parameters in
the Vigo estuary but at different stations and frequency.

In the summer of 1984, the *Galicia VIII* cruise studied the summer upwelling events
occurring on the contact front between the two ENACW water masses off Cape
Finisterre from short sections perpendicular to the Galician coast with 85 stations
offshore and 35 stations over the shelf. This cruise marked a milestone in the
oceanographic research of IIM-CSIC because it was the first time that the parameters of
the carbon system were measured on-board in offshore waters. Moreover,
measurements of a particular station on the shelf break with a bottom depth of 600
metres were taken every two days for a month, including two-day continuous
samplings.

Two years later, the Ria de Vigo 1986 sampled along the main axis of the Ria de Vigo
in 7 monthly repetitions during the first half of the year in which the primary production
and the organic matter exchange between the estuary and the shelf was studied in
relation to the hydrographic regime. Shortly afterwards, the same topic was also
researched by the Galicia IX project in September and October 1986 from 145 stations,
50 of which were coastal and 80 located in ocean waters (Prego et al., 1990).

The following year, the 1987 Provigo project (Nogueira et al., 1997) initiated a periodic
study from a fixed site (42º14.5'N, 8º45.8'W) located in the main channel in the middle
zone of the *Ría de Vigo*. This oceanographic station was selected as suitable for
evaluating the main processes that occur in the inner ria associated with external forcing
changes (Rios, 1992; Figueiras et al., 1994). Although the Provigo project finished in
1996, the fixed station was repeatedly included in subsequent cruises, extending the
time series at this location until today, when it is currently sampled every week by
INTECMAR (www.intecmar.gal). An example of the subsequent sampling repetition of
this station occurred the following year when one of the three stations in the Vigo
estuary in the Luna 1988 project (Fraga et al., 1992) took a sample every two weeks to

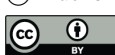



study the environmental control over the phytoplankton populations throughout an
annual cycle (February 1988 - February 1989).

At the end of 80s, the carbon system monitoring by the IIM-CSIC was extended to the
*Ría de Arousa* throughout 1989 (Álvarez-Salgado et al., 1993; Perez et al., 2000) in
order to learn the effect of upwelling on the water circulation pattern, community
production and the fluxes and net budgets of biogenic constituents in this ria with the
highest mussel production in Europe. For 5 months, 11 stations' samples were repeated
twice a week in the ria that is the most productive, housing intense cultivation of
mussels on rafts (Blanton et al., 1984).

**Cruises of the 90s:**
In the first half of this decade, studying the phytoplankton communities was the
oceanographic cruises' most relevant aim, concentrating particularly on harmful algae
blooms. The hydrodynamic and biogeochemical conditions controlling the growth,
development and migration of the phytoplankton were analysed both in the interior of
the estuary and on the continental shelf.

For five days in September, the 1990 *Ría de Vigo* cruise (Figueiras et al., 1994) sampled
five stations distributed along the longitudinal axis of the ria and one at the northern
mouth. The next year, the cruise Galicia XI was carried out in May, sampling at 39
stations along eight transects perpendicular to the coastline; and Galicia XII (Alvarez-
Salgado et al., 1998, 2002, 2003; Castro et al., 1994) in September, sampling at 37
oceanic stations and 7 coastal stations.

The *Ría de Vigo* cruise in 1993-4 (Miguez et al., 2001), with four stations using 24
repetitions with a CTD-SBE25, investigated the hydrodynamic and biogeochemical
effect on the evolution of phytoplankton communities in the *Ría de Vigo*. Six samples
were taken in approximately two weeks corresponding to two different periods
(27 September to 8 October 1993, and 6 March to 24 March 1994).

*Ría de Vigo* 1994-95 (Alvarez et al., 1999; Doval et al., 1998, 1997a, 1997b) and *Ría de*
*Vigo* 1997 (Gago et al., 2003a, b, c) were two cruises that took place in the second half
of the decade. These campaigns' objective was no longer the ecology of the plankton,



but the factors behind the variation of the carbon pools during the upwelling and
downwelling events along the central axis of the *Ría of Vigo*. During the 1997 cruises
on board the *B/O Mytilus*, a systematic observation of the $pCO_2$ was carried out for the
first time in Spanish coastal waters, using an autonomous continuous system with
additional measurements of temperature, salinity and chlorophyll.

**209 Cruises in the 2000s and recent years:**

After a period of poor sampling at the end of 90s, the first decade of the 21[st] century
gave new impetus to biogeochemical monitoring of Galician waters. As shown below,
several projects dealt with various objectives, focussing on particular issues in the
dynamics of these waters:

The DYBAGA project (Galician Platform's Annual Dynamics and Biochemistry: short-
scale variation) (Álvarez-Salgado et al., 2006; Castro et al., 2006; Nieto-Cid et al.,
2004) analysed the phenomena of upwelling and downwelling in the Galician shelf
opposite the *Ría de Vigo* weekly and their impact on the different biogeochemical and
carbon system variables including organic dissolved matter. Three stations were
sampled weekly from May 2001 to April 2002 between the shelf break (1,200 m deep)
to the middle of the *Ría de Vigo* (45 m deep).

The REMODA (Reactivity of dissolved organic matter in a coastal upwelling system)
(Álvarez-Salgado et al., 2005; Piedracoba et al., 2005; Nieto-Cid et al., 2006) project
concentrated on learning the origin and destination of dissolved organic matter in the
*Ría de Vigo* as well. Three stations along the main axis of the *Ría de Vigo*, including the
fixed station as the central one, took samples with short (3-4 days) and seasonal time
scales.

The FLUVBE project (Coupling of benthic and pelagic fluxes in the *Ría de Vigo*) added
to knowledge about the productivity and the benthic fluxes of oxygen and inorganic
nutrients in the Ría de Vigo from 16 oceanographic surveys with four stations between
April 2004 and January 2005.

The ZOTRACOS project studied the biogeochemical and hydrodynamic
characterization of the coastal transition zone in NW Spain during the downwelling
period (Teira et al., 2009).

The CRIA (Circulation in a RIA) (Barton et al., 2019) project examined the layout of
the two-layer circulation and propagation of upwelled and downwelled waters in order
to estimate the flushing and vertical velocities in the *Ría de Vigo* in repeated
hydrographic surveys between September 2006 and June 2007 (Barton et al., 2015,
2016; Alonso-Perez and Castro, 2014; Alonso-Perez et al., 2010; Alonso-Perez et al.,
244    2015).


The RAFTING project (Impact of mussel raft cultivation on the benthic-pelagic
coupling in a Galician ria) (Frojan et al., 2018; Frojan et al., 2016; Froján et al., 2014)
assessed for the first time how mussel cultivation influences the quality of particular
organic carbon fluxes in the *Ría de Vigo*. Over the four seasons, two stations were
visited every two to three days during each period, meaning 24 oceanographic cruises in
2007 and 2008.

The CAIBEX (Continental shelf-ocean exchanges in the marine ecosystem of the
Canary Islands-Iberian Peninsula) (Villacieros-Robineau et al., 2019) project compared
the dynamics and biogeochemical activity between the coastal zone and the adjacent
ocean in the study zone during the summer upwelling events. As part of the CAIBEX
project, a mooring at the LOCO (Laboratory of Ocean and Coastal Observation)
(Zuñiga et al., 2016, 2017) site located on the continental shelf was deployed and visited
monthly for one year to monitor the vertical profiles of biogeochemical variables.

After these projects were completed in 2009, new measurements were not provided
until 2018. The aim of the ARIOS project (Acidification in the rias and on the Iberian
continental shelf) was to evaluate the impact of ocean acidification and learn about
potential impacts on the mussels and their adaptation (Lassoued et al., 2019) to the new
climate change.

**3.3. Methods**





To assess of the level of acidification in the ocean adjacent to the Galician coast,
variables of the carbon system (pH and alkalinity), nutrient concentration, dissolved
oxygen, chlorophyll-a, salinity and temperature were measured in each cruise. The
variables measured in each oceanographic cruise gathered in the ARIOS dataset are
shown in Table 1. The main changes in the material and methods throughout these years
are detailed below.

**T-S measurements**
Temperatures from 1976 to 1984 were measured using a Wallace and Tiernan
bathythermograph. Reversing thermometers were used, attached to the water samplers
between 1984 and 1990, correcting the temperature between the protected and
unprotected thermometers according to Anderson (1974). During those years, the depth
was calculated from the thermometric readings, rounding the result off to the nearest
ten. After 1990, different models of CTD instruments often containing other sensors
were used to obtain the thermohaline profile.

The first measurements of salinity were determined with a Plessey Environmental
Systems 6230N inductive salinometer calibrated with normal IAPSO water and
calculated from the equations given in the NIO and UNESCO International
Oceanographic Tables (1981). After using this equipment, the salinity was determined
with an AUTOSAL 8400A inductive salinometer calibrated with normal IAPSO water
whose estimated analytical error was 0.003, using the equations given by UNESCO
(1981) as well. CTDs began to be used in 1990 to create the vertical salinity profiles,
calibrated using the salinity samples, whose possible deviations in the measurements
were estimated from the discrete measurements from the AUTOSAL salinometer.

**pH measurements**
The pH measurements were originally taken with a Metrohm E-510 pH meter with a
glass electrode and a Ag/ClAg reference one calibrated with 7.413 NBS buffer. All pH
values were converted to values at 15 ºC using the temperature correction from the
Buch and Nynas tables published by Barnes (1959). In 1984, the method was modified
and the temperature normalization was carried out following Pérez and Fraga (1987b).
Two years later, the measurement equipment was the Metrohm E-654 pH meter with an
Orion 81-04 Ross combined glass electrode, with the pH converted to the SWS scale



using the hydrogen activity coefficient given by Mehrbach et al. (1973) at 25ºC with the
parameterization given by Pérez and Fraga (1987b). The error in this potentiometric
method was 0.010. In 2001, the seawater pH measurements were determined with a
spectrophotometric method following Clayton and Byrne (1993), subsequently adding
0.0047 to the pH value to do so (DelValls and Dickson, 1998). The precision of the
spectrophotometric measurements was 0.003 pH units.

The pH values were reported on total pH scale at 0 dbar of pressure and both at 25ºC
and in situ temperature following the same procedure of GLODAP v2 (Olsen et al.,
2019). A total of 12,220 measurements of pH on NBS scale were converted to the total
scale using CO2SYS (Lewis and Wallace, 1998) for MATLAB (van Heuven et al.,
2011) with pH and total alkalinity as inputs. The conversion was conducted with the
carbonate dissociation constants of Lueker et al. (2000) and the borate-to-salinity ratio
of Uppström (1974). Whenever total alkalinity data were missing, these values were
approximated as 66 times salinity that is the mean ratio between the total alkalinity and
the salinity of every in situ measurements compiled in the ARIOS database. Data for
phosphate and silicate are also needed and were, whenever missing, a constant values of
10 µmol kg$^{-1}$ for silicate and 1 µmol kg$^{-1}$ for phosphate were used. These
approximations were tested on 8,296 samples with complete biogeochemical
information showing a bias of less than 0.0004 pH units for 99.95% of the samples.

**Alkalinity measurements**
The seawater alkalinity was measured for the first time in 1981 by potentiometric
titration with HCl 0.1 N at final pH 4.44 following Pérez and Fraga (1987a) with an
analytical error of 2 µmol kg$^{-1}$ and a precision of 0.1%. Sodium tetraborate decahydrate
(Borax, Na$_2$B$_4$O$_7$ 10H$_2$0, Merck p.a.) was used for standardizing the HCl (0.13 M). The
pH measurements were carried out with a combined glass electrode (Metrohm E-121)
with Ag/AgCl (KC1 3M) as the reference. The pH was calibrated using the NBS buffers
assuming the theoretical slope. As of 2001, the accuracy of alkalinity measurements
was determined using samples of certified reference material (CRM) provided by Dr. A.
Dickson, University of California, improving the precision to ±1.4 mol kg$^{-1}$ and an
accuracy of <0.1% recently established by (Ríos and Pérez, 1999) from cross-
calculation with measured Certified Reference Materials (Dickson et al., 2007).



**Nutrient measurements**


Except in the campaigns called Galicia, where the samples were analysed on board, the
samples collected are kept in the dark and cold (4ºC) to be analysed in the laboratory.
Nutrient concentration was determined by a flow-segmented autoanalyzer (Technicon
AAII and Alpkem after 1995) as described in Strickland and Parsons (1968) with the
particularity that the reduction of nitrate to nitrite with Cd column is done using a citrate
buffer according to Mouriño and Fraga's modification (1985). Phosphates and silicates
were measured following Grasshoff (1983), and ammonium as described by Grasshoff
and Johannsen (1972). This method was maintained in the subsequent cruises,
achieving a precision of 0.02 µmol/kg for nitrite, 0.1 µmol/kg for nitrate, 0.05 µmol kg$^{-1}$
for ammonium and silicate, and 0.01 µmol/kg for phosphate.

**Oxygen measurements**


The dissolved oxygen was determined via the Winkler titration method for the first time
in 1981 following the procedure published later by Culberson et al. (1991). The oxygen
concentration in the samples in this method is fixed with $Cl_2Mn$ and NaOH/NaI, which
are kept in the dark until analysis in the laboratory 12-24 hours later. The measurements
were made by titration of iodine with thiosulfate using an automatic titrator. During the
80s and early 90s, the titration was carried out with Metrohm instruments (E-425 or E-
473), which had an analytical error of 1 µmol kg$^{-1}$. The oxygen concentration after 1997
was estimated using a Titrino 720 (Metrohm) analyser with an accuracy of 0.5 µmol kg$^{-1}$

357 .


**Chlorophyll measurements**


The chlorophyll-a values were measured following SCOR-UNESCO (1966) using a
6 cm diameter Schleicher and Scholl 602eh filter covered with magnesium carbonate.
The absorption was measured in 1 cm optical path cuvettes using a Beckman DU
spectrophotometer. In 1984, discrete water samples of the chlorophyll-a samples were
filtered through Whatman GF/C filters of 2.5 cm, which were preferred from then on,
and measured fluorometrically following Strickland and Parsons (1972) without
correction for concentration by pheophytes. The fluorescence readings were carried out
with a Turner Designs 10,000 R fluorometer (Yentsh and Menzel, 1963) obtaining a
precision of 0.05 g L$^{-1}$.




**Quality control**

Every cruise gathered in the Table 1 passed 1st quality control to ensure truly confident
results. The GO-SHIP software for quality control of hydrographic data (Velo et al.,
2019) that compile several QC procedures was applied to ARIOS dataset. A quality flag
was assigned to each measurement available from the repository sites (Table 2). This
method was preferred over applying a very stringent flagging process because it is
difficult to rule out some extreme values associated with low salinities or that could be
supported by the high variability of an ecosystem with very high biological activity

The ARIOS database includes the cruise corrections for pH data of the -0.017 for the
Galicia VIII cruise (29GD19840711) and +0.032 for Galicia IX cruise
(29GD19860904) detected during the second level quality control of CARINA project
(Velo et al., 2009).

**3.4. Distribution of sampling**

According to the type of region under study, different areas were identified in order to
classify the measurements gathered in the oceanographic cruises (Fig. 1). The latitude
of 43ºN where Cape Finisterre is located was used as the dividing line between northern
and southern waters. Subsequently, a criterion of depth also split the waters to the north
of 43ºN into north oceanic (below 250 m), north shelf (between 205 m and 75 m) and
north coast (75 m to the surface). The southern shelf waters were divided by latitude
42ºN into Portuguese and the *Rías Baixas* (RB) shelves, whereas the shallower waters
were identified by the main rias, where three different zones were defined using
longitude boundaries (outer, middle and inner) according to Gago et al. (2003c) in the
*Ría de Vigo*, and just two zones in the other rias (*Ría de Pontevedra*, *Ría de Arousa*, *Ría*
*de Muros*). Southern waters between the isobath at 75 metres and the mouth of the
estuaries were identified as the Portuguese and RB coast.

The discrete measurements gathered in the ARIOS dataset were mainly found in
different regions' waters around 42ºN latitude (Fig 1; Fig. 2a), especially in the outer
and middle areas of the *Ría de Vigo*, which accounted for 15% and 21% of the total
measurements respectively due to the proximity to the *Instituto de Investigaciones*
*Mariñas* (IIM-CSIC). Most of the measurements (85%) carried out by many of these



cruises to study the coastal ecosystems concentrated on shallow waters between the
seawater surface and 75 metres in depth (Fig. 2b). Although waters below 4,900 metres
deep were also sampled, observations below 900 metres only account for 1% of the
ARIOS database.

The observations made over more than 40 years in every region of the ARIOS database
were irregular on both an interannual and seasonal scale (Fig. 2a). The period of most
sampling activity was the 80s and 90s, whereas samples were especially scarce in the
early 2010s. On a seasonal scale, summer and autumn were the preferred seasons to
address the different research purposes, with 37% and 36% of the total samples
respectively. The observations taken during less favourable winter conditions,
especially aboard the coastal vessels usually available, only accounted for the 10% of
the ARIOS database.

**4. Results**
Some of the most obvious results provided by the ARIOS database are shown below.
The purpose is to describe the environmental context and the main oceanographic
processes that affect the variability of these discrete measurements and offer
preliminary information for future detailed biogeochemical research.

**4.1. Vertical distribution**
The vertical profile in the ocean region between 41ºN and 43ºN was estimated for each
oceanographic station as the mean value of the depth ranges described in Figure 2b.
These measurements were gathered attending to the collection periods (December-
February, March-May, June-August and September-November) and averaged to
describe winter, spring, summer and autumn respectively (Fig. 3).

The vertical distribution of the temperature (Fig. 3a) showed the presence of warmer
saline waters throughout the water column in winter with the exception of the surface
waters during summer, which showed intense heating due to the radiant solar energy.
Below the maximum temperature observed during the summer, cold central waters of
subpolar origin occupied the water columns with lower salinity (Fig. 3b). The vertical
variation of temperature is typical for a temperate region with relatively homogenous
deep water below the seasonal thermocline, reaching maximum SST values in summer



and autumn, and minimums in spring and winter. The winter temperature profile is
relatively warmer than in spring because of the presence of the IPC (Alvarez-Salgado et
al., 2006), which reaches a depth of 300 metres. The maximum salinity is also found in
winter due to the presence of the IPC, whereas the minimum values are found in autumn
(Fig. 3b). Below 500 metres in depth, the increase in salinity points to the presence of
Mediterranean Water. These differences reach a minimum at 500 metres deep, where
the salinity values coincided. From this depth to 1,100 metres, the differences in
temperature and salinity throughout the four seasons were minimal, with the mean
values converging to 11.03±0.07°C and 36.117±0.009 psu, respectively (Fig. 3ab).

The vertical distribution of pH, $NO_3^-$ and oxygen concentration (Fig. 3cde) also showed
a variation lower than 1% at this depth with annual means of 15.2±0.1 $\mu$mol kg$^{-1}$,
8.025±0.005 and 188±1 $\mu$mol kg$^{-1}$ respectively. The pH values from a maximum
subsurface located at around 40 metres deep showed a clear inverse correlation with the
depth down to a depth of 500 metres throughout the seasonal cycle, where the annual
minimum value of 8.018±0.005 was reached. The highest values were related to the
biological $CO_2$ drawdown, which brought the pH to a peak value of 8.13 at 40 metres
deep during the spring bloom. Underneath this intense photosynthesis activity between
the surface and 100 metres, the respiration of organic matter took the pH to lower
values than those measured in winter between 200 to 500 metres, a depth at which the
spring and winter values were practically equal. The impact on the growth of the
phytoplankton community during the spring was also evident, judging by the oxygen
concentration. So, in the upper waters the spring oxygen concentration values exceeded
those of the winter values, while oxygen consumption was found from a depth of
300 metres to 1,000 metres due to respiration from organic matter arriving from above.
The minimum values for oxygen concentration throughout the water column were found
during summer and autumn. The nitrate concentration displayed a particularly vertical
distribution, growing with depth from minimum values in the upper layer of the ocean
region, which was practically zero during the first 50 metres. Below 100 metres, the
nitrate concentration showed the maximum values in the vertical distribution during
summer and autumn coinciding with the presence of waters of subpolar and subtropical
origin respectively, whereas the minimum values appeared in winter. Towards the
bottom, the seasonal values of $NO_3^-$ concentration were almost coincident at a mean



value of 15.2±0.1 μmol kg$^{-1}$.

**4.2. Seasonal cycle**
The seasonal cycle of the surface waters (0 to 5 metres) was estimated as a monthly
filtered means, accepting only values within two standard deviations. Five regions that
were located as a longitudinal transect between the inner *Ría de Vigo* and the ocean
zone are shown in Fig. 4.

In general terms, the seasonal variability of the temperature was very similar in every
area, ranging between 12 and 19ºC (Fig. 4a). Only particular features observed on a
short-term scale as in the examples below differ between each region. The warmer
waters were usually found in the oceanic zone, reaching a maximum monthly averaged
temperature of 18.6ºC in September, while the coldest surface waters of 12.6ºC were
located in the inner stations closer to the mouth of the *Ría de Vigo* in January. Another
secondary minimum averaged temperature was also found in the shelf and the outer area
of the *Ría de Vigo*, which was remarkably low in August due to the entry of cold
upwelled waters in the surface layer (Alvarez-Salgado 1993).

The monthly salinity averages (Fig. 4b) clearly showed significant differences between
the offshore and coastal waters. Sharp salinity changes were seen in the estuary during
winter, especially in the inner area where values lower than 28 psu were reached with
the arrival of continental inputs in December. The weak seasonal cycle of salinity in the
shelf and ocean waters showed high values in December due to the influence of warm
saline water from the IPC, usually located on the shelf slope even though it may even
enter the rias depending on the relative intensity of shelf winds and the intensity of the
continental runoff (Alvarez-Salgado et al., 2003). In this sense, the slight salinity
minimum observed in the shelf waters in March could be consequence of the offshore
spreading of the maximum discharges from the River Miño and Douro (Otero et al.,
2010) at the end of downwelling season. After this, the shelf and ocean waters showed
minimum values in summer due to the arrival of cooler and fresher subpolar waters
(Rios et al., 1992; Alvarez-Salgado et al., 2003, 2006). In August, coinciding with the
maximum salinity of the surface waters in the interior of the *Ría de Vigo* due to the
minimum river runoff, the surface waters between the inner *Ría de Vigo* and the ocean
region were almost homogeneous, with minimum differences in salinity of 0.2 psu.




Like salinity, there was little seasonal variability in pH in the offshore waters, but large
seasonable variability in coastal waters, with peak pH in spring and minimum pH in
autumn in every region (Fig. 4c). The net balance between production and respiration of
organic matter and the estuarine circulation caused a maximum pH of 8.19 in the outer
region of the *Ría de Vigo* in May and a minimum of 7.96 in the inner waters in
November.

The oxygen concentration (Fig. 4d) in the coastal ecosystems is also controlled by the
remineralization of the organic matter and photosynthetic activity of the phytoplankton
community, with the effect of salinity and temperature on the oxygen saturation level.
The variability in the oxygen concentration, like the pH distribution, showed a growing
seasonal amplitude towards the coastline, with maximum values in the outer and middle
*Ría de Vigo* and lower values in the inner waters, especially during the second half of
the seasonal cycle. Hence, the dissolved oxygen concentration mirrored the seasonal
cycle of pH, showing growing seasonal amplitude towards the coastline with a range
between 284 μmol kg$^{-1}$ found in the outer region of the *Ría de Vigo* in May and 205
μmol kg$^{-1}$ in the inner waters in November. These results seem to reinforce the
importance of the oxygen consumption in this shallow area, where the water column is
less than 10 metres deep and so it would also be influenced by benthic respiration
(Alonso-Pérez and Castro, 2014).

The monthly means of nitrate concentration (Fig. 4e) could be summarized as high
values during autumn and winter due to the nutrients delivered from the continent and
the vertical mixing, and as minimum nitrate values from March to September because of
phytoplankton consumption. The nitrate concentration was markedly higher in the inner
*Ría de Vigo*, where it exceeded 9 μmol kg$^{-1}$ in February and decreased towards the open
ocean, where the highest monthly value was seen to be 2.5 μmol kg$^{-1}$. Some notable
aspects can be seen in Fig. 5d, such as water poor in nitrate in the ocean region between
the two peaks of 3.5 μmol kg$^{-1}$ in March and 1.3 μmol kg$^{-1}$ in October. This shows the
presence of the IPC waters, which are warmer and saltier than the shelf waters. Also
noteworthy was the particular fact that while the nitrate concentration in other areas was
practically zero in summer, the nitrate amount in the surface waters within the *Ría de*



*Vigo*, and especially in the inner *Ría de Vigo*, was not completely consumed. This
indicates a constant supply throughout the year, either through upwelling events or the
continental inputs. This in turn means that while the chlorophyll values were at a
minimum in the offshore waters in summer, the phytoplankton community in the
estuary grew in summer during the upwelling relaxation periods (Pérez et al., 2000).
The nutrient concentration during spring and summer is only detectable in the recently
upwelled waters. It can reach up to 6 μmol L$^{-1}$ (Fraga, 1981; Castro et al., 1994). During
the cessation of the upwelling season in September and October, the chlorophyll
concentration (Fig. 5f) rises again, sustained by nutrients entering from deeper waters
through vertical mixing. It should be noted that there is a coincidence of high
chlorophyll in the water column and low oxygen concentration in the inner *Ría de Vigo*
from May to November, indicating the potential importance of benthic fluxes and
vertical fluxes.

**4.3. Long-term trend**
The long-term trends of these surface waters were estimated to be the interannual linear
rate of the deseasonalyzed time series, previously removing the monthly means in these
regions and assuming a null spatial variability. The significant trends in the ARIOS
database, meaning long-term variability, should be interpreted as a combination of the
natural variability on a decadal scale (Pérez et al., 2010; Padin et al., 2010) and
anthropogenic forcings (Wolf-Gladrow et al., 1999; Anderson and Mackenzie 2004;
Bakun et al., 2010).

No long-term temperature variability was found in the surface waters of any region
despite the known warming previously reported on the Northern Iberian coast (Pérez et
al., 2010; Gesteira et al., 2011; González-Pola et al., 2005). Unlike the temperature, the
other expected consequence of climate change (Caldeira and Wickett, 2003), namely
ocean acidification was observed along the longitudinal transect, with a greater decline
in pH number towards the coast (Table 2). The long-term pH variation of -
0.0039±0.0005 yr$^{-1}$ in the inner waters was about triple the change of -
0.0012±0.0002 yr$^{-1}$ in the ocean zone, explaining the 34% and 22% variation in pH in
situ respectively, and representing 1-3% of the seasonal pH variation in all zones. Other
acidification rates estimated in different sites of the North Atlantic Ocean (Lauvset and
Gruber, 2014; Bates et al., 2014) including -0.018 decade$^{-1}$ in the mean global ocean pH



(Lauvset et al., 2015) or -0.0164 decade[-1] in the Eastern North Atlantic by Ríos et al.
(2001) are within the acidification range found in the ocean and coastal zones of these
waters.

The long-term trend in salinity was also seen to be evidently dependent on the distance
to the mouth of the ria. The interannual rate of sea surface salinity in the outer and inner
*Ría de Vigo* previously reported by Rosón et al. (2009) was 0.0426±0.016 psu yr[-1] and
0.0193±0.0056 psu yr[-1] respectively. These changes were observed in parallel to an
interannual alkalinity increase that is cancelled out in the normalized alkalinity,
estimated as the difference between the alkalinity measured and the alkalinity calculated
using the linear regression with salinity in each region. So, the interannual salinity
increase was the forcing that explains the increase in the buffer capacity of the surface
waters (Sarmiento and Gruber, 2006).

Other significant long-term variations were found in other biogeochemical parameters
in the ARIOS database. The long-term trend of the nutrient concentration in the inner
*Ría de Vigo* showed an increase in the nitrate, phosphate and ammonium concentrations
of        0.0158±0.006 μmol kg[-1] yr[-1],        0.0076±0.0016 μmol kg[-1] yr[-1]        and
0.0560±0.0011 μmol kg[-1] yr[-1] respectively (Doval et al., 2016). This fertilization on a
long-term scale in the surface waters of the inner ria was observed in parallel to the
deoxygenation of -0.7±0.2 μmol kg[-1] yr[-1]. The apparent oxygen utilisation (AOU),
calculated using the concentration of $O_2$ at saturation calculated according to Benson
and Krause (1984), underwent an equivalent significant long-term change of
0.7±0.2 μmol kg[-1] yr[-1], indicating that either the biological consumption rates, or a
change in the amount of time that the waters are ventilated, or even its interaction or
exchange with the sediment, cause the the long-term reduction of oxygen.  .

Attending to the interannual salinity changes in the shallower waters of the *Ría de Vigo*,
these findings seem to be related to a weakening of the vertical salinity gradient and a
growing exchange between the bottom and surface waters. So, the footprint of oxygen
consumption and the remineralized nutrient inputs resulting from benthic respiration
seem to reach the upper layer. The metabolic processes in a more homogeneous water
column would also explain an intense acidification in the inner waters in spite of
growing alkalinity buffering.




The mean values at each station of the ARIOS database estimated for each depth range
described in Figure 2, resulting in 8,384 values, were used to estimate a general value of
the long-term trend in pH. The historical pH values in situ from the ARIOS database
showed a general decrease in seawater pH in the Iberian Upwelling between 1976 and
2018, with an acidification rate of $-0.012\pm0.002$ $yr^{-1}$ that significantly explains 2% of
the total pH variation (Fig. 5a). The apparent oxygen utilisation was also shown as
function of pH over time, revealing the association of higher AOU values with lower
pH. The relationship between pH and AOU (Fig. 5b) showed an inverse linear
correlation of $-399\pm5$ $\mu mol$ $kg^{-1}$. The strong biological activity of the upwelling systems
is the main driver of pH changes, explaining 52% of the observed variation in the
discrete measurements. The distribution of nitrate seen in relation to the distribution of
pH and AOU (Fig. 5b) showed the association of higher pH values with negative AOU
values and a nitrate  decrease, reinforcing the importance of biological processes in
these marine carbonate system.

Although the different processes controlling the AOU values were not separated in this
analysis, the oxygen concentration in addition to the remineralization and
photosynthesis of organic matter is conditioned by changes in temperature and salinity,
ventilation events, water masses mixing and other processes (Sarmiento and Gruber,
2006). Therefore, the long-term drop in seawater pH measurements in this region
responding to the intrusion of atmospheric $CO_2$ may also be due to the impact of other
interannual changes affecting the seawater pH, such as biological activity. So, the future
evolution of acidification will respond both to the potential increase in $CO_2$ in the
atmosphere and to other long-term changes affecting the seawater's carbonate system.

**5. Data availability**
The ARIOS dataset (Pérez et al., 2020) is archived at DIGICAL CSIC under the Digital
Object Identifier (DOI): http://dx.doi.org/10.20350/digitalCSIC/12498.

The data are available as WHP-Exchange bottle format (arios_database_hy1.csv). A
documentation file (readme_ARIOSDATABASE.txt) provides an description of the
material and methods of the measurements and the parameters of the dataset. In both
files, a table similar to the Table 1 of this manuscript include the DOI and the



639 EXPOCODE of the original cruise files gathered in the ARIOS dataset.


641 These data are available to the public and the scientific community with the aim of that

642 their wide dissemination will lead to new scientific knowledge about the ocean

643 acidification and the biogeochemistry of the Galicia Upwelling System. The dataset is

644 subject to a Creative Commons License Attribution-ShareAlike 4.0 International

645 (http://creativecommons.org/licenses/by-sa/4.0/) and users of the ARIOS dataset should

646 reference this work.

647

648 **6. Conclusions**

649 The ARIOS database is a unique compilation of biogeochemical discrete measurements

650 in the Iberian Upwelling Ecosystem from 1976 to 2018. This data set comprises more

651 than 17,653 discrete samples from 3,357 oceanographic stations (but not always for all

652 parameters) of pH, alkalinity and associated physical and biogeochemical parameters

653 (e.g., temperature, salinity, and chlorophyll and oxygen concentrations). The material

654 and methods varied throughout the sampling period due to logistical and analytical

655 issues such as those described in Table 1, where different sites are mentioned to

656 download these measurements and detailed information.

657

658 Among the results described as preliminary and relevant information to learn the

659 environmental and oceanographic context of the ARIOS database, we can mention the

660 following main points concerning the pH characteristics of the Iberian Upwelling

661 System:

662  • A decrease in seawater pH in the Iberian Upwelling between 1976 and 2018,

663   with an acidification rate of $-0.012\pm0.002$ yr$^{-1}$ that significantly explains 2% of

664   the total pH variation

665  • An interannual pH variation of $-0.0039\pm0.0005$ yr$^{-1}$ in the inner waters and $-$

666   $0.0012\pm0.0002$ yr$^{-1}$ in the ocean zone.

667  • An inverse linear correlation between pH and AOU of $-399\pm5$ $\mu$mol kg$^{-1}$ that

668   explained 52% of the observed variation in the discrete measurements.


670 This published ARIOS database is a useful and necesseary tool to confirm and study

671 biogeochemical changes in the seawater at long term trend. Likewise, we understand

672 that it is a starting point to which to add future observation projects to continue





increasing the knowledge about the impact of climate change in the Iberian Upwelling
Ecosystem.

**Acknowledgements.**
The compilation of this data set was funded by the ARIOS project (CTM2016-76146-
C3-1-R) funded by the Spanish government through the Ministerio de Economía y
Competitividad that included European FEDER funds. Part of the processing work was
supported by the MarRISK project (European Union FEDER 0262_MarRISK_1_E)
funded by the Galicia-Northern Portugal Cross-Border Cooperation Program
(POCTEP). This project has also received funding from the European Union's Horizon
2020 research and innovation programme under grant agreement No 820989 (project
COMFORT, Our common future ocean in the Earth system – quantifying coupled
cycles of carbon, oxygen, and nutrients for determining and achieving safe operating
spaces with respect to tipping points). This data set encompasses decades of work
conducted by an overwhelming number of people. We thank all of the scientists,
technicians, personnel, and crew who were responsible for the collection and analysis of
the over 22 000 samples included in the final data set. In addition to the PI cited in
Table 1 we also thank to Trinidad Rellán, Antòn Velo, Miguel Gil Coto, Marta Alvarez,
Marylo Doval, Jesus Gago, Daniel Broullón and Marcos Fontela. We also thank Monica
Castaño for starting this data compilation more than 10 years ago.

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

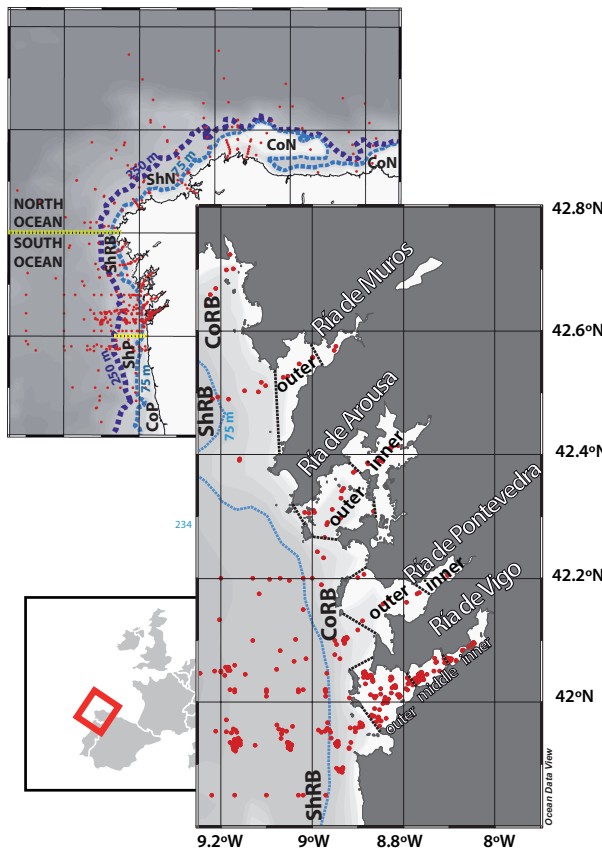

Figure 1. Map of all stations (red dots) including the geographical areas selected to
classify the ARIOS database from isobath of 250 m (dark blue line) and 75 metres (light
blue line), latitudinal criterion (yellow lines) and geographical lines (black lines).





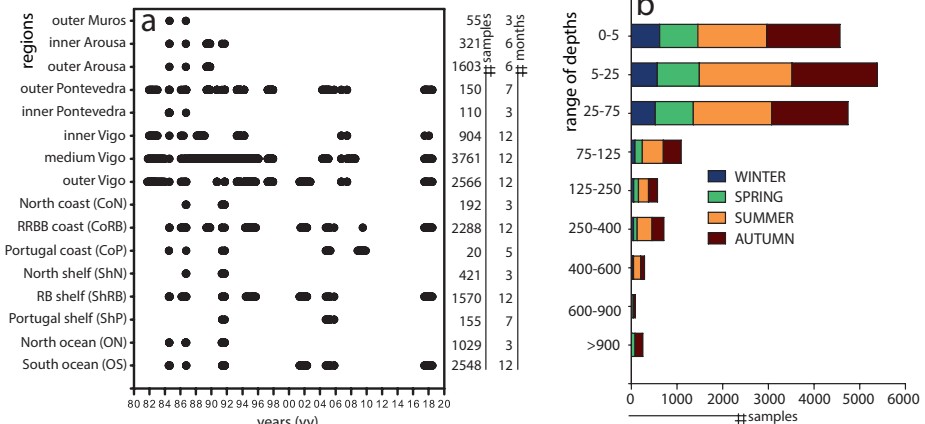



Figure 2. a) Temporal distribution of the observations in the geographical boxes
included in the ARIOS dataset. b) Seasonal distribution of the measurements in relation
to depth.




















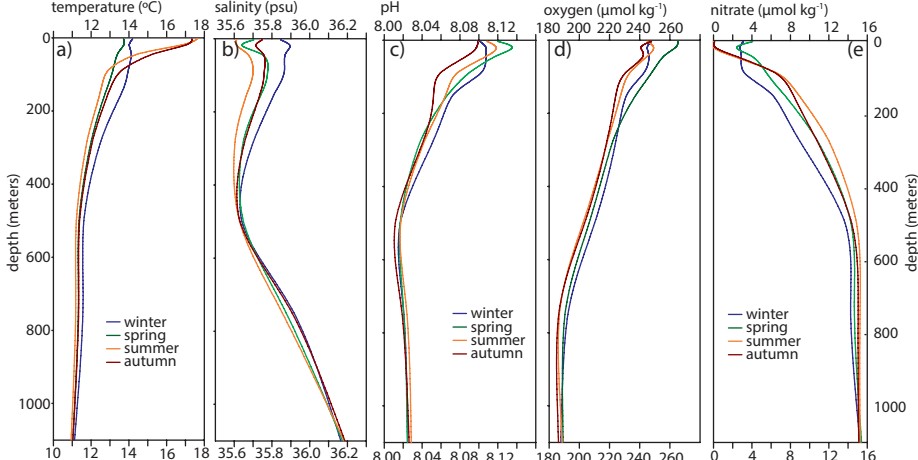



Figure 3. Profiles of seasonal means of temperature (a), salinity (b), pH (c), oxygen (d)
and nitrate concentration (e) in the first 1100 meters of the region South Ocean shown
in Fig. 1.






















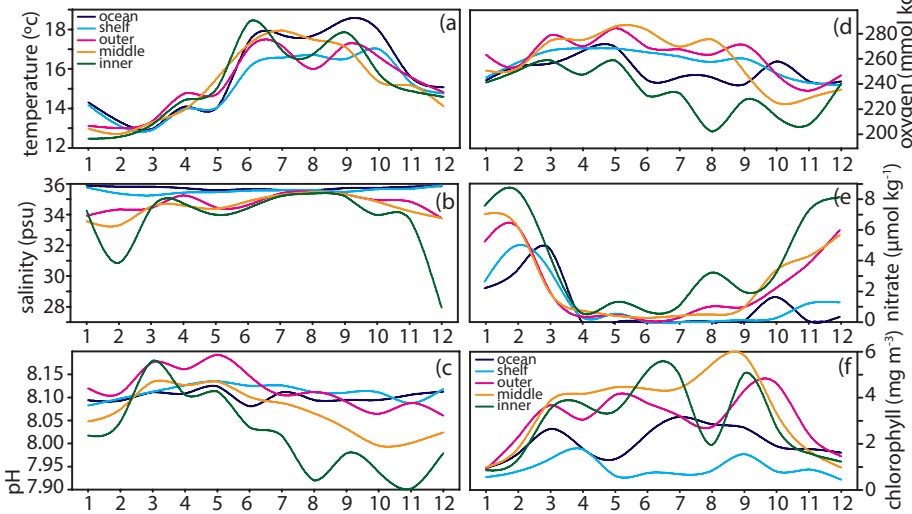



Figure 4. Sea surface (<5 meters depth) seasonal cycles in 1976 - 2018 of temperature
(a), salinity (b), pH (c), oxygen concentration (d), nitrate concentration (e) and
chlorophyll  (f) at sea surface for five geographical boxes shown in Fig. 1: South Ocean,
RB shelf and outer, middle and inner Ria de Vigo for the entire period of the ARIOS
database.



















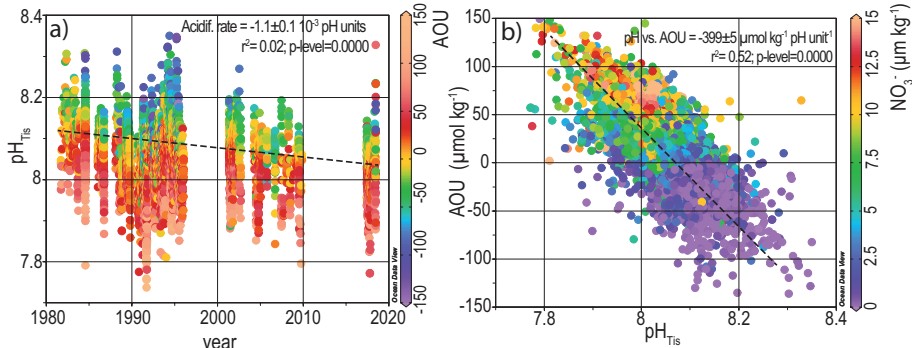


Figure 5. Time-series of pH ARIOS data. The black line depicts the long-term trend.
Scatter diagram of AOU vs pH including the nitrate concentration shown as colour of
every dot.




















| EXPOCODE | PROJECT | DATE | IP | # | CTD | $O_2$ | Nut | pH | Alk | Chla | CRM | Data Repository | REGIONS |
|---|---|---|---|---|---|---|---|---|---|---|---|---|---|
| 29LP19761026 | Ria Vigo 1977 | 1976-10-26 | F Fraga | 135 | N | N | S* | S° | N | N | N | http://dx.doi.org/10.20350/digitalCSIC/9917 | $Co^{RB}$ |
| 29LP19810929 | Ria Vigo 1981-83 | 1981-09-29 | F Fraga | 748 | N | N | S* | S° | S | N | N | http://dx.doi.org/10.20350/digitalCSIC/9918 | $RV^{O,M,I}$ |
| 29LP19830215 | Ria Vigo 1983-84 | 1983-02-15 | F Fraga | 312 | N | S* | S* | S° | S | N | N | http://dx.doi.org/10.20350/digitalCSIC/9919 | $RV^{O,M}$ |
| 29GD19840711 | GALICIA-VIII | 1984-07-11 | F Fraga | 1865 | N | S | S | S° | S | S | N | http://dx.doi.org/10.20350/digitalCSIC/9908 | $O^{N,S}$, $Sh^{RB}$, $Co^{P,RB}$, $RV^{O,M,I}$, $RA^{O,I}$, $RP^{O,I}$, RM |
| 29GD19860121 | Ria Vigo 1986 | 1986-01-21 | F Fraga | 332 | N | S | S | S° | S | S | N | http://dx.doi.org/10.20350/digitalCSIC/9910 | $Sh^{RB}$, $Co^{RB}$, $RV^{O,M,I}$ |
| 29GD19860904 | GALICIA-IX | 1986-09-04 | F Fraga | 1640 | N | S | S | S° | S | S | N | http://dx.doi.org/10.20350/digitalCSIC/9911 | $O^{N,S}$, $Sh^{RB,N}$, $Co^{P,RB,N}$, $RV^{O,M,I}$, $RA^{O,I}$, $RP^{O,I}$, RM |
| 29LP19870120 | PROVIGO | 1987-01-20 | FF Pérez | 2317 | N | S | S | S° | N | S | N | http://dx.doi.org/10.20350/digitalCSIC/9924 | $RV^{M}$ |
| 29LP19880212 | LUNA 88 | 1988-02-12 | AF Rios | 468 | N | S | S | S° | S | S | N | http://dx.doi.org/10.20350/digitalCSIC/9907 | $RV^{M,I}$ |
| 29IN19890512 | GALICIA-X | 1989-05-12 | FF Pérez | 3113 | N | S | S | S° | S | S | N | http://dx.doi.org/10.20350/digitalCSIC/9920 | $Co^{RB}$, $RA^{O,I}$ |
| 29IN19900914 | Ria Vigo 1990 | 1990-09-14 | FG Figueiras | 108 | Y | S | S | S° | S | S | N | http://dx.doi.org/10.20350/digitalCSIC/9921 | $RV^{O,M,I}$ |
| 29IN19910510 | GALICIA-XI | 1991-05-10 | FF Pérez | 327 | Y | S | S | S° | S | S | N | http://dx.doi.org/10.20350/digitalCSIC/9922 | $O^{N,S}$, $Sh^{P,RB,N}$, $Co^{P,RB,N}$, $RA^{O}$ |
| 29IN19910910 | GALICIA-XII | 1991-09-10 | FG Figueiras | 663 | Y | S | S | S° | S | S | N | http://dx.doi.org/10.20350/digitalCSIC/9923 | $O^{N,S}$, $Sh^{P,RB,N}$, $Co^{P,RB,N}$, $RV^{O,M,I}$, $RA^{O}$ |
| 29LP19930413 | Ria Vigo 1993-94 | 1993-04-13 | FG Figueiras | 406 | Y | S | S | S° | S | S | N | http://dx.doi.org/10.20350/digitalCSIC/9927 | $RV^{O,M,I}$ |
| 29JN19940505 | Ria Vigo 1994-95 | 1994-05-05 | M Cabanas | 669 | Y | S | S | S° | S | S | N | http://dx.doi.org/10.20350/digitalCSIC/9926 | $Sh^{RB}$, $Co^{RB}$, $RV^{O}$ |
| 29MY19970407 | CIRCA-97 | 1997-04-07 | FF Pérez | 547 | Y | S | N | S° | S | S | N | http://dx.doi.org/10.20350/digitalCSIC/9928 | $RV^{O,M,I}$ |
| 29MY20010515 | DYBAGA | 2001-05-15 | FF Pérez | 1421 | Y | S | S* | S | S | S | Y | http://dx.doi.org/10.20350/digitalCSIC/9929 | $Sh^{P,RB}$, $Co^{RB}$, $RV^{O}$ |
| 29MY20010702 | REMODA | 2001-07-02 | XA Alvarez | 203 | Y | S | S* | S | S | S | Y | http://dx.doi.org/10.20350/digitalCSIC/9930 | $RV^{O}$ |
| 29MY20040419 | FLUVBE | 2004-04-19 | CG Castro | 187 | Y | S | S* | S | S | S | Y | to be submitted | $RV^{M,I}$ |
| 29CS20041004 | ZOTRACOS | 2004-10-04 | M Cabanas | 371 | Y | S | S | S | S | S | Y | http://dx.doi.org/10.20350/digitalCSIC/9932 | $Sh^{P,RB}$, $Co^{P,RB}$, $RP^{O}$ |
| 29MY20060926 | CRÍA | 2006-09-26 | D Barton | 197 | Y | S | S* | S | S | S | Y | http://dx.doi.org/10.20350/digitalCSIC/9931 | $RV^{O,M,I}$ |
| 29MY20070917 | RAFTING | 2007-09-17 | CG Castro | 287 | Y | S | S* | S | S | S | Y | to be submitted | $RV^{M}$ |
| 29MY20081105 | LOCO | 2008-11-05 | XA Alvarez | 72 | Y | S | S* | S | S | S | Y | http://dx.doi.org/10.20350/digitalCSIC/9936 | $Co^{RB}$ |
| 29AH20090710 | CAIBEX-I | 2009-07-10 | D Barton | 191 | Y | S | S | S | S | S | Y | http://dx.doi.org/10.20350/digitalCSIC/9934 | $Co^{P,RB}$ |
| 29MY20170609 | ARIOS | 2017-06-09 | FF Pérez | 1114 | Y | S | S* | S | S | S | Y | http://dx.doi.org/10.20350/digitalCSIC/9963 | $Sh^{P,RB}$, $Co^{RB}$, $RV^{O,M,I}$ |



Table 1. Discrete measurements of projects gathered in the ARIOS database and
associated information: including dates, sample number (#), the principal investigator
(PI), measured parameters, link to data repository and the sampled geographical area.

*- All projects include measurements of T, S. Others as pH, alkalinity (Alk), nutrient*
*(Nut), oxygen ($O_2$) concentration, chlorophyll (Chla) are indicated.*
*- The concentration units of these variables are µmol kg$^{-1}$ or µmol L$^{-1}$ (\*) and the pH*
*measurements in NBS scale (°) or in total scale.*
*- Regions are identified as ocean (O), shelf (Sh), coastal (Co), Ría de Vigo (RV), Ría de*
*Pontevedra (RP), Ría de Arousa (RA) and Ría de Muros (RM) while the superscript*
*index means south ($^S$), north ($^N$), Portugal ($^P$), Rías Baixas ($^{RB}$), outer ($^O$), middle ($^M$)*
*and inner ($^I$).*

























|         | $SS_{range}$ | $r^2ss$ | $t_{interannual}$ | $r^2$ | p-value |
|---------|--------------|---------|-------------------|-------|---------|
| OCEAN   | 0.050        | 0.17    | -0.0012±0.0002    | 0.21  | 0.0000  |
| SHELF   | 0.050        | 0.06    | -0.0017±0.0003    | 0.15  | 0.0009  |
| OUTER   | 0.120        | 0.24    | -0.0027±0.0003    | 0.21  | 0.0000  |
| MIDDLE  | 0.130        | 0.28    | -0.0022±0.0005    | 0.03  | 0.0000  |
| INNER   | 0.260        | 0.47    | -0.0039±0.0005    | 0.34  | 0.0000  |


Table 2: Seasonal amplitude of monthly pH means (SSrange) and long-term trends
($t_{interannual}$) of pH in five regions and significant regression coefficients between the in
situ pH measurements and the monthly mean pH values ($r^2ss$) and the regression
coefficient of the temporal variability of the deseasonalyzed pH measurements ($r^2$).



