# Peer review of "ARIOS: a database for ocean acidification assessment in the Iberian Upwelling"

_Earth System Science Data, 2020_

## Referee Comment (RC1) · Anonymous Referee #1 · 9 May 2020

The paper by Padin et al. describes a detailed data base of carbon system parameters in the North Western Iberian Peninsula collected during 32 years. Not many of these datasets exist from upwelling areas and particularly in coastal regions, which makes this dataset very valuable. The authors performed an analysis to show seasonal cycles and long term trends of the biogeochemical parameters considered. Ocean acidification rates in surface waters of the Iberian Upwelling System are also provided. Finally, the study gives insights on the processes responsible for the temporal changes observed in the different parameters.

The manuscript is scientiïfically sound. With regard to the presentation, the paper

is easily readable although some sentences are unclear, and there are a number of grammatical errors and typos in the text. If the authors have their text double checked, I only have some small remarks which need to be dealt with. Therefore, overall, I believe that the work is well suited for publication in ESSD.

Specific comments:

Title should be modified. I would suggest something like ARIOS: a database for ocean acidification assessment in the Iberian Upwelling System.

Introduction Line 41: change "fix" to "withdraw" Lines 42-45: please rephrase "In any case, the rapid increase of CO2 in the atmosphere decreases the ocean's pH". I would suggest: The gradual absorption of atmospheric CO2 by the oceans decreases seawater pH, causing ocean acidification, which conditions the buffering capacity of seawater and in turn the exchange of CO2 between the ocean and the atmosphere. Lines 52-53: Please rephrase. "observe and gather data about pH and other parameters of the marine carbon system to produce global and regional data products in order to help sustainably manage the ocean's resources. I would suggest: to conduct accurate measurements of pH and ancillary parameters and provide data products for a sustainable management of marine resources. Line 55: change of marine ecosystems to for marine ecosystems Lines 63-65: I know what you mean but I would recommend to rephrase the sentence. Line 67-72: remove the s at the end of effects for grammar consistency. I would also move this paragraph after line 53 for coherence with the text. Lines 72-75: I would modify the entire paragraph as: In the Iberian Upwelling System, accurate measurements of carbon system parameters commenced more than 30 years ago. Lines 75-80: which changes? Results have not been presented yet. I would therefore, continue the statement as: Researchers of the Instituto de Investigaciones Mariñas (IIM-CSIC) have been collecting pH and biogeochemical data along the Galicia coast (40°N and 45°N, 11°W) under the framework of different projects. This has allowed to generate a database, ARIOS (Acidification in the rias and the Iberian continental shelf), containing 17,653 discrete records gathered in 3,357 sampling sites.

Data provenance: I would remove provenance Line 86: I would replace Region by Data Coverage Line 115: change In addition to for Besides Line 119: delete the article before acidification and it would be convenient to specify the exact region/s where the mentioned acidification rate was estimated

Data sources: In general, I very much appreciated the comprehensive explanation of the projects that provided data for the dataset. However, considering that explanatory information of the cruises is given in Table 1 and each individual project is associated to a database included in a public repository, I do not find section 3.2 essential for the manuscript, as all those details can be mentioned (and possible are) in the repository. The authors might re-consider to shorten this section by keeping the first paragraph and refer subsequent info to Table 1.

Methods: Lines 337-338: Please rephrase. I would suggest: Except for the Galicia cruises (Table 1), in which nutrient samples were analysed on board, samples were kept in the dark and cold (4°C) after collection for further analyses in the shore based laboratory. Line 341: change is to was. Line 351: Same as above Line 371: remove the article before Table 1 Line 374: You possibly mean Table 1 instead? Line 377: by the high variability present in a system characterized by an intense biological activity Line 384: This section should be moved and either merged with 3.1 or placed right below it for the sake of consistency and for a better introduction of the sampling region.

Results: Line 424: vertical profile of what? Please indicate. Lines 420-445: why don't you show the standard deviations for T and S for all the depth ranges as you do for the water column comprised between 500m and 1100m? Line 443: add …down to 1100m Line 447: change distribution to profile Line 448: replace at this depth by within this depth range Lines 452-454: Speculative as it is not demonstrated or shown in the graph. Therefore, I would just say: The highest pH values could be attributed to the biological CO2 drawdown by phytoplankton activity, which brought the pH to a peak value of 8.13 at 40 metres deep during the spring bloom. Lines 454-457. Same as above. I would suggest to rephrase the paragraph as it is also confusing. Below 100

[Figure]

metres, respiration of organic matter possibly was responsible of lowering pH. . ..but anyhow the text s counterintuitive To me, pH values between 200 and 500 m depth seem to be lower than those from 500 m down to 1100m, which were also constant and similar within the entire depth range regardless of the season. Lines 458-462: Please rewrite: I would propose: The influence of phytoplankton growth on biogeochemistry during spring can be also evidenced by the oxygen concentration pattern during this season. In the upper layer (depth range?) spring oxygen levels exceeded those in winter, whereas a decrease in oxygen concentration was found from 300 m depth down to 1000 m, possible due to enhanced respiration from cascading organic matter.

It would be helpful to add in this section a table with averaged concentrations and SD of each parameter within the different depth ranges and for each season.

Line 473: seasonal cycle of what? Please specify. I would recommend to rewrite the whole paragraph, as in Fig 4 what you actually show is the seasonal cycle of different biogeochemical parameters in surface waters of 5 regions and not the five regions themselves, as it can be deduced from the text the way it is right now.

Line 506: replace seasonable by seasonal.. you could also rewrite the following sentence as: with maximum and minimum pH values in spring and autumn, respectively, and in all regions (Fig. 4c).

Lines 541-549: considering change to past tense for consistency with the rest of the paragraph. Moreover, a reference could be well added at the end of the paragraph to reinforce your statement regarding the relevance of benthic and vertical fluxes in the Ria.

Line 551: please add a "s" to trend Line 552: long term trends of what? Please specify. I assume the temporal trends are estimated over parameters and they do not refer to surface waters themselves. Therefore, it needs to be re-written. Line 562: any suggestion why a warming trend is not found as it was previously reported? Line 563: . . .consequence of climate change. . .I would add "in marine ecosystems".. and move

the reference by Cladeira and Wicket 2003 after ocean acidification. Line 565: What do you mean by pH number? Value? Number of measurements? Line 566: please replace . . .was about triple the change of. . .by. . ... was three fold higher than the trend observed in the open ocean zone, equivalent to 0.0012±0.0002 yr-1 Line 569-575: I would rephrase the paragraph as: These pH decrease rates found in both coastal and open ocean regions of the Iberian Upwelling System lie within the range of other acidification rates estimated in different sites of the North Atlantic Ocean (Lauvset and Gruber, 2014; Bates et al., 2014), being also coherent with the mean rates calculated for the global ocean and for the Eastern North Atlantic and equal to -0.018 and -0.0164 decade-1 , respectively (Lauvset et al., 2015; Rios et al 2001)

Line 575: just indicate: Salinity exhibited an increasing long-term trend (value?) that was dependent on the distance to the mouth of the Ria (de Vigo? All of them?). I do not see the salinity trend indicated anywhere. Line 581: Change So to Therefore. Line 586: add a "s" to nutrient and remove the previous article Line 587: same as above: delete the article before nitrate. But anyhow, it is not clear if the trends in nutrients level come from the previous study by Doval et al (2016) or are the result of your analysis. Please clarify. Lines 590-596: do you mean that your AOU temporal trend coincides with the deoxygenation rate calculated previously by Doval? It is not clear enough in the text. Lines 598-604: Speculative. Please support with references Line 614: correlation coefficient should be indicated even though it is contained in the Figure. Line 606-619: To me, the entire paragraph is the highlight of the paper, as it evidences the relevance of the dataset and gives insight on the processes responsible for the mean decreasing pH trend found in the area. In my opinion, this finding gets somehow diluted between the other results when it should be emphasized by the authors. Line 623: photosynthesis of organic matter??? Line 627: I would finish as: Hence, the analysis performed over the database presented here confirms that the future evolution of ocean acidification in this productive region is likely to depend on both the potential CO2 increase in the atmosphere and other long-term changes (of natural and/or anthropogenic origin) affecting the seawater's carbonate system.

---

## Referee Comment (RC2) · Michele Giani (Referee) · 5 Jun 2020

The data of pH and total alkalinity, dissolved oxygen, nutrients and chlorophyll together with the main physical variables is impressive as it is based on measurement on 17653 water samples, comprising more than 4 decades, from 1976 to 2018. Notwithstanding the methods changed over the decades, most of the information is sufficient to evaluate the QA/QC of the measurements. The vertical distribution, seasonal variability and of the temporal trends of the $CO_2$ system are presented and help to evaluate the quality and the usefulness of these measurements. The data set surely will be relevant also for the future understanding of the interactive effects on acidification in the

Iberian Upwelling System of coastal processes and global changes. However there is a need of a careful revision of the data set and of improvements in the ms. In the abstract the author give acidification rates ranging from – 0.0016 to -0.0032 pH units/yr whereas in the ms (L. 566) they give a -0.0039 ph units/yr for the inner waters. This discrepancy should be resolved. As the estimated acidification rate is higher than the average ocean acidification it would be important to discuss the potential effects of the gaps in the times series some spanning also 7 consecutive years. It would be relevant to compare the trends on periods without long gaps of data which could strongly affect the slope of the trend. A comparison with other articles reporting ranges for coastal acidification trends could be interesting for improving the discussion of results. In the section "Cruises in the 2000s and recent years" (L. 261- 265), the information about each cruise is given but not always the months and years are given. I strongly suggest to provide similar information for each cruise or to refer to a more specific table where the time span of each cruise is given. In particular regarding the last ARIOS project it is not clear in which months was carried out. It would be important if the authors could be the precision for the temperature and salinity measurements in the period 1976-1984. For chlorophyll measurements, as different filters were used, could the authors provide an estimation of the pore size given and of potential effects of the change. The indication of the volume filtered (range) could be also important if available. Regarding the adopted Quality control procedure (L.370-L.382) it would be useful if the authors could provide a synthetic information on the first and second level of the quality control cited in this section. Regarding the presented ARIOS data set there are some corrections to the data that the authors should consider as there are many negative concentrations for nitrites (n=4), nitrates (n=16), ammonia (n=13), and chlorophyll a (n=2). There are concentrations for nutrients and chlorophyll a in the range of 10-3 to <10-7 that should be correctly reported, presumably, as less than the detection limits given in the methods, and properly flagged with QF = 6. For all nutrients there are many values equal to 0 with QF= 2, these values presumably are below the detection limits and should be flagged with QF=6. There are three in situ pH values in the range

7-7.6 that should be checked to evaluate if they can be considered reliable or doubtful. Below some minor comments are given: L.52 I suggest to correct as follows: to help a sustainable management of the . . . L. 290 I think that "as well" should be omitted and I suggest to substitute "to create" with "to record". L. 306 "pH value to do so": unclear. L325-327 it is unclear if for the titration the HCl concentration was 0.1 or 0.13 M. L.338, L.341 change the conjugation of verb to the past. L.351 "Cl2Mn" should be written as "MnCl2". L.352 change the conjugation of verb to the past. L. 382 check the year in the reference list is 2010. L. 506 "large seasonable variability" change with " large seasonal variability". L. 523 "and so it would" I suggest changing as: "and therefore it would". I suggest changing the yellow colour in Figure 1, as on the printed version is not clearly visible. To enhance the readability of Figure 2, I suggest to enlarge them or to split the figure in two. In Figures 3 and 4, for Salinity, I suggest omitting "psu" as it is not a real measurement unit, but a conductivity ratio. I suggest to indicate the pH is on the total scale similarly to figure 5. There is the need to correct the units of oxygen in micromole kg-1 in figure 4. The subscript of pHT in the Figure 5 is not well readable and should be explained in the caption. I suggest, all over the figures, to indicate the pH as pHT for clarifying that is expressed on the total scale, moreover, to increase the readability, I suggest to enlarge or split the two graphs. TABLE 1. According to the data set, the ARIOS cruises were carried out during different months of 2018 and not in one month of 2017. As some cruises/projects could span over more month perhaps it would be better to provide the period of the study instead of a single date.

---

## Author Comment (AC1) · 24 Jul 2020

The manuscript is scientifically sound. With regard to the presentation, the paper is easily readable although some sentences are unclear, and there are a number of grammatical errors and typos in the text. If the authors have their text double-checked, I only have some small remarks, which need to be dealt with. Therefore, overall, I believe that the work is well suited for publication in ESSD.

Specific comments:
Title should be modified. I would suggest something like ARIOS: a database for ocean acidification assessment in the Iberian Upwelling System.
**The title has been changed to the following:**
**ARIOS: a database for ocean acidification assessment in the Iberian Upwelling System (1976 - 2018).**

**Introduction**
Line 41: change "fix" to "withdraw"
**The suggestion has been included in the new version of the manuscript.**

Lines 42-45: please rephrase "In any case, the rapid increase of CO2 in the atmosphere decreases the ocean's pH". I would suggest: The gradual absorption of atmospheric CO2 by the oceans decreases seawater pH, causing ocean acidification, which conditions the buffering capacity of seawater and in turn the exchange of CO2 between the ocean and the atmosphere.
**The suggestion has been included in the new version of the manuscript**

Lines 52-53: Please rephrase. "observe and gather data about pH and other parameters of the marine carbon system to produce global and regional data products in order to help sustainably manage the ocean's resources. I would suggest: to conduct accurate measurements of pH and ancillary parameters and provide data products for a sustainable management of marine resources.
**The suggestion has been included in the new version of the manuscript**

Line 55: change of marine ecosystems to for marine ecosystems
**The suggestion has been included in the new version of the manuscript**

Lines 63-65: I know what you mean but I would recommend to rephrase the sentence.
**The sentence "However, the great spatial and temporal variability has been widely spaced and intermittent over time, preventing a complete view from being obtained of ocean acidification in the upwelling system" has been rewritten. The new sentence is: "However, the high physical/chemical variability in short temporal and spatial scales of upwelling systems and the lack of regular sampling in these waters prevents a complete picture of the acidification of these ecosystems."**

Line 67-72: remove the s at the end of effects for grammar consistency. I would also move this paragraph after line 53 for coherence with the text.
**The following paragraph has been moved after line 53.**
**"The effect of ocean acidification on marine ecosystems has stimulated impetus in the international community for gathering high quality time-series measurements of the marine inorganic carbon system (Hofmann et al., 2011; Andersson and MacKenzie, 2012; McElhany and Busch, 2013; Takeshita et al., 2015; Wahl et al., 2016) and for predicting the future evolution of the pH caused by climate change."**

Lines 72-75: I would modify the entire paragraph as: In the Iberian Upwelling System, accurate measurements of carbon system parameters commenced more than 30 years ago.
**The first sentence of the new paragraph is the following "In the Iberian Upwelling System, the researchers of the Instituto de Investigaciones Mariñas (IIM-CSIC) since 1976 commenced accurate measurements of marine inorganic carbon system and associated parameters. As a result, a collection of pH observations and ancillary biogeochemical information along the Galicia coast (40ºN and 45ºN, 11ºW) has been gathered under the framework of different projects over the past 40 years. The current database, hereinafter called ARIOS (Acidification in the rias and the Iberian continental shelf) database, holds biogeochemical information from 3,357 oceanographic stations, giving 17,653 discrete samples."**

Lines 75-80: which changes? Results have not been presented yet. I would therefore, continue the statement as: Researchers of the Instituto de Investigaciones Mariñas (IIM-CSIC) have been collecting pH and biogeochemical data along the Galicia coast (40_N and 45_N, 11_W) under the framework of different projects. This has allowed to generate a database, ARIOS (Acidification in the rias and the Iberian continental shelf), containing 17,653 discrete records gathered in 3,357 sampling sites.
**In response to the previous two comments, the paragraph has been rewritten in order to include the suggestions: "In the Iberian Upwelling System, the researchers of the Instituto de Investigaciones Mariñas (IIM-CSIC) since 1976 commenced accurate measurements of marine inorganic carbon system and associated parameters. As a result, a collection of pH observations and ancillary biogeochemical information along the Galicia coast (40ºN and 45ºN, 11ºW) has been gathered under the framework of different projects over the past 40 years."**

**Data provenance:** I would remove provenance
Line 86: I would replace Region by Data Coverage
**Region has been replaced by Data spatial coverage**

Line 115: change In addition to for Besides
**The suggestion has been included in the new version of the manuscript**

Line 119: delete the article before acidification and it would be convenient to specify the exact region/s where the mentioned acidification rate was estimated
**The suggestion has been included and the region was mentioned as "the acidification in the first 700 metres for the geographical area from the Iberian Peninsula to the 20º W meridian and from 36ºN to 43ºN has also been observed at a rate of -0.0164 pH units per decade"**

**Data sources:**

In general, I very much appreciated the comprehensive explanation of the projects that provided data for the dataset. However, considering that explanatory information of the cruises is given in Table 1 and each individual project is associated to a database included in a public repository, I do not find section 3.2 essential for the manuscript, as all those details can be mentioned (and possible are) in the repository. The authors might re-consider to shorten this section by keeping the first paragraph and refer subsequent info to Table 1.

**We understand that detailed information about the methods and materials of the oceanographic cruises that are part of the ARIOS database is mandatory according to the requirements of ESSD. Therefore, we prefer not to modify this section of the manuscript.**

**Methods:**

Lines 337-338: Please rephrase. I would suggest: Except for the Galicia cruises (Table 1), in which nutrient samples were analysed on board, samples were kept in the dark and cold (4_C) after collection for further analyses in the shore based laboratory.

**The suggestion has been included in the new version of the manuscript**

Line 341: change is to was.

**The suggestion has been included in the new version of the manuscript**

Line 351: Same as above

**The suggestion has been included in the new version of the manuscript**

Line 371: remove the article before Table 1

**The suggestion has been included in the new version of the manuscript**

Line 374: You possibly mean Table 1 instead?

**The suggestion has been included in the new version of the manuscript**

Line 377: by the high variability present in a system characterized by an intense biological activity

**The suggestion has been included in the new version of the manuscript**

Line 384: This section should be moved and either merged with 3.1 or placed right below it for the sake of consistency and for a better introduction of the sampling region.

**This section has been moved just below section 3.1. The other sections of the Material and Methods have been renumbered because of this change.**

**Results:**

Line 424: vertical profile of what? Please indicate.

**These sentences had been rewritten. The new text is: "The vertical profile of the temperature, salinity, $pH_T$, $NO_3^-$ and oxygen concentration in the ocean region between 41ºN and 43ºN was estimated for each oceanographic station as the mean value of the depth ranges described in Figure 2b"**

Lines 420-445: why do not you show the standard deviations for T and S for all the depth ranges as you do for the water column comprised between 500m and 1100m?

**The standard deviations were not included in Figure 3 simply to facilitate the clarity of the graphical representation of the vertical biogeochemical profiles. For the same reason, they were also not included in the monthly distribution of the biogeochemical variables shown in Figure 4. In any case, the mean values with their corresponding standard deviation values (mean±standard deviation) represented in Figure 3 and Figure 4 are shown in two Tables at the end of the reply to the referee. This information could be included as supplementary material to the manuscript if it deems this appropriate by the Editor.**

Line 443: add : : :down to 1100m
**The suggestion has been included in the new version of the manuscript**

Line 447: change distribution to profile
**The suggestion has been included in the new version of the manuscript**

Line 448: replace at this depth by within this depth range
**The suggestion has been included in the new version of the manuscript**

Lines 452-454: Speculative as it is not demonstrated or shown in the graph. Therefore, I would just say: The highest pH values could be attributed to the biological CO2 drawdown by phytoplankton activity, which brought the pH to a peak value of 8.13 at 40 metres deep during the spring bloom.
**The suggestion has been included in the new version of the manuscript.**

Lines 454-457. Same as above. I would suggest to rephrase the paragraph as it is also confusing. Below 100 metres, respiration of organic matter possibly was responsible of lowering pH: : :.but anyhow the text s counterintuitive To me, pH values between 200 and 500 m depth seem to be lower than those from 500 m down to 1100m, which were also constant and similar within the entire depth range regardless of the season.
**The commented lines: "Underneath this intense photosynthesis activity between the surface and 100 metres, the respiration of organic matter took the pH to lower values than those measured in winter between 200 to 500 metres, a depth at which the spring and winter values were practically equal." were rewritten. This was the new text: "The higher pH values could be attributed to the biological reduction of $CO_2$ by phytoplankton activity, which brought the pH to a maximum value of 8.13 to 40 meters during the spring bloom. After the intense photosynthetic activity observed in surface waters during spring and summer, pH values reached minimum values in the first 200 meters of depth during autumn due to respiration of organic matter. However, it was at a depth of 500 metres that the minimum pH values were measured in all seasons where is found the subpolar Eastern North Atlantic Central Water proceeding from the northeastern cyclonic gyre (Harvey, 1982; Ríos et al., 1992)."**

Lines 458-462: Please rewrite: I would propose: The influence of phytoplankton growth on biogeochemistry during spring can be also evidenced by the oxygen concentration pattern during this season. In the upper layer (depth range?) spring oxygen levels exceeded those in winter, whereas a decrease in oxygen concentration was found from 300 m depth down to 1000 m, possible due to enhanced respiration from cascading organic matter. It would be helpful to add in this section a table with averaged concentrations and SD of each parameter within the different depth ranges and for each

season.

**The suggestion has been included in the new version of the manuscript. The new lines are the following: The influence of phytoplankton growth on biogeochemistry during spring can be also evidenced by the oxygen concentration pattern (Fig. 3e). In the upper layer above 250 metres depth, spring oxygen levels exceeded those in winter, whereas a decrease in oxygen concentration was found from this depth down to 1000 metres, possible due to enhanced respiration from cascading organic matter.**

Line 473: seasonal cycle of what? Please specify. I would recommend to rewrite the whole paragraph, as in Fig 4 what you actually show is the seasonal cycle of different biogeochemical parameters in surface waters of 5 regions and not the five regions themselves, as it can be deduced from the text the way it is right now.

**The suggestion has been included in the new version of the manuscript. The new lines are the following: "The seasonal cycle of the biogeochemical properties (temperature, salinity, pH, oxygen concentration, nitrate concentration and chlorophyll) in the surface waters (0 to 5 metres) of five geographical boxes was estimated as a monthly average previously filtering values outside of two standard deviations of the mean. Five regions that were located as a longitudinal transect between the inner Ría de Vigo and the ocean zone are shown in Fig. 4."**

Line 506: replace seasonable by seasonal.. you could also rewrite the following sentence as: with maximum and minimum pH values in spring and autumn, respectively, and in all regions (Fig. 4c).

**The suggestion has been included in the new version of the manuscript**

Lines 541-549: considering change to past tense for consistency with the rest of the paragraph. Moreover, a reference could be well added at the end of the paragraph to reinforce your statement regarding the relevance of benthic and vertical fluxes in the Ria.

**The suggestion has been included in the new version of the manuscript. The new lines are the following: "The nutrient concentration during spring and summer was only detectable in the newly upwelled waters that can show values up to 6 $\mu$mol L$^{-1}$ (Fraga, 1981; Castro et al., 1994). During the cessation of the upwelling season in September and October, the chlorophyll concentration (Fig. 5f) increased again, sustained by nutrients that entered from deeper waters through vertical mixing. It should be noted that there was a coincidence of high chlorophyll in the water column and low oxygen concentration in the inner *Ría de Vigo* from May to November, indicating the potential importance of benthic fluxes and vertical fluxes (reference).**

Line 551: please add a "s" to trend

**The suggestion has been included in the new version of the manuscript**

Line 552: long term trends of what? Please specify. I assume the temporal trends are estimated over parameters and they do not refer to surface waters themselves. Therefore, it needs to be re-written.

**The text has been re-written. "The long-term trends of the biogeochemical properties in these surface waters were estimated to be the interannual linear rate of the deseasonalyzed time series, previously removing the monthly means in these**

**regions and assuming a null spatial variability"**

Line 562: any suggestion why a warming trend is not found as it was previously reported?
**The fact that we did not find a statistically significant warming trend as would be expected on the basis of the overall ocean behavior is mainly due to the fact that the warming trend was estimated for the surface waters of the study area. The first meters of the water column in this coastal zone are under the influence of important oceanographic phenomena that affect temperature such as coastal upwelling pulses, the presence of surface currents or river inputs. The different temporal variability of these processes together with the remarkable spatial variability of the study area prevents the characterization of the warming in a statistically significant way from measurements with the irregular frequency of the ARIOS database. Therefore, information from other temperature data sources and a more detailed statistical analysis should be considered for this specific purpose.**

Line 563: consequence of climate change: : :I would add "in marine ecosystems".. and move the reference by Caldeira and Wicket 2003 after ocean acidification.
**The suggestion has been included in the new version of the manuscript**

Line 565: What do you mean by pH number? Value? Number of measurements?
**The text has been re-written. "with a greater decline in pH number towards the coast" has been changed by "with a greater decrease in the long-term trend of pH towards the coast**

Line 566: please replace : : :was about triple the change of: : :by: : :.. was three fold higher than the trend observed in the open ocean zone, equivalent to 0.0012_0.0002 yr-1
**The suggestion has been included in the new version of the manuscript**

Line 569-575: I would rephrase the paragraph as: These pH decrease rates found in both coastal and open ocean regions of the Iberian Upwelling System lie within the range of other acidification rates estimated in different sites of the North Atlantic Ocean (Lauvset and Gruber, 2014; Bates et al., 2014), being also coherent with the mean rates calculated for the global ocean and for the Eastern North Atlantic and equal to -0.018 and -0.0164 decade-1 , respectively (Lauvset et al., 2015; Rios et al 2001)
**The suggestion has been included in the new version of the manuscript**

Line 575: just indicate: Salinity exhibited an increasing long-term trend (value?) that was dependent on the distance to the mouth of the Ria (de Vigo? All of them?). I do not see the salinity trend indicated anywhere.
**The suggestion has been included in the new version of the manuscript**

Line 581: Change So to Therefore.
**The suggestion has been included in the new version of the manuscript**

Line 586: add a "s" to nutrient and remove the previous article
**The suggestion has been included in the new version of the manuscript**

Line 587: same as above: delete the article before nitrate. But anyhow, it is not clear if the trends in nutrients levelcome from the previous study by Doval et al (2016) or are the result of your analysis. Please clarify.

**The rates had been estimated from ARIOS database. This paragraph has been rewritten in order to clarify the meaning. The new text is the following: " The long-term trend of the concentrations of nutrients in the inner Ría de Vigo that had been previously reported for the period 2001-2011 by Doval et al. (2016) showed a significant increase in nitrate, phosphate and ammonium concentrations of 0.0559±0.0158 µmol kg-1 yr-1, 0.0076±0.0016 µmol kg-1 yr-1 and 0.0560±0.0011 µmol kg-1 yr-1 respectively."**

Lines 590-596: do you mean that your AOU temporal trend coincides with the deoxygenation rate calculated previously by Doval? It is not clear enough in the text.

**Long-term trends in oxygen concentration and AOU were estimated from the ARIOS database. The paragraph was rewritten to clarify this information. This is the new text: "This fertilization on a long-term scale estimated from ARIOS database in the surface waters of the inner ria was observed in parallel to the deoxygenation of -0.7±0.2 µmol kg-1 yr-1. The apparent oxygen utilisation (AOU), calculated using the concentration of oxygen at saturation calculated according to Benson and Krause (1984), underwent a long-term change of 0.7±0.2 µmol kg-1 yr-1 equal to the observated in the measurements of oxygen concentration. This coincidence may indicates that the long-term reduction of oxygen is due to the changes in the biological consumption rates, in the rates of the waters ventilation or even in sediment-water interactions rather than due to the effect of temperature and salinity on oxygen saturation.**

Lines 598-604: Speculative. Please support with references

**This paragraph was intended to provide an integrative hypothesis on observed large-scale trends. The text has been changed in the manuscript to the following paragraph: "These findings found in the shallower waters of the *Ría de Vigo* allow us to hypothesize that the long-term increase in salinity would produce an increasingly weak vertical salinity gradient in the water column that would favour the vertical fluxes between the bottom and surface waters. Therefore the observed changes of oxygen and remineralized nutrient inputs in the surface waters could be due to an increasing footprint of benthic respiration, that has a major importance in the net ecosystem metabolism of this coastal region (Alonso-Pérez et al., 2015). This hypothesis would also explain the intense acidification in the inner waters in spite of growing alkalinity buffering."**

Line 614: correlation coefficient should be indicated even though it is contained in the Figure.

**The coefficient of determination of 0.52 between pH and AOU was included in the sentence as follows: "The relationship between pH and AOU (Fig. 5b) showed an inverse linear correlation of -399±5 $\mu$mol kg$^{-1}$ and a coefficient of determination (r-squared) of 0.52."**

Line 606-619: To me, the entire paragraph is the highlight of the paper, as it evidences the relevance of the dataset and gives insight on the processes responsible for the mean decreasing pH trend found in the area. In my opinion, this finding gets somehow diluted between the other results when it should be emphasized by the authors.

Line 623: photosynthesis of organic matter???
**The sentence has been corrected. The new text is: "... the oxygen concentration in addition to the remineralization of the organic matter and the photosynthesis is conditioned by changes..."**

Line 627: I would finish as: Hence, the analysis performed over the database presented here confirms that the future evolution of ocean acidification in this productive region is likely to depend on both the potential CO2 increase in the atmosphere and other long-term changes (of natural and/or anthropogenic origin) affecting the seawater's carbonate system.
**Following the suggestion, the new manuscript finish as follows: "Therefore, the long-term drop in seawater pH measurements estimated from the ARIOS database presented here confirms that the future evolution of ocean acidification in this productive region is likely to depend on both the potential CO2 increase in the atmosphere and other long-term changes (of natural and/or anthropogenic origin) affecting the seawater's carbonate system."**

[revised manuscript text omitted]

---

## Author Comment (AC2) · 24 Jul 2020

The data set surely will be relevant also for the future understanding of the interactive effects on acidification in the Iberian Upwelling System of coastal processes and global changes. However there is a need of a careful revision of the data set and of improvements in the ms. In the abstract the author give acidification rates ranging from – 0.0016 to -0.0032 pH units/yr whereas in the ms (L. 566) they give a -0.0039 ph units/yr for the inner waters. This discrepancy should be resolved. **The discrepancy has been corrected. The correct acidification rate is -0.0039 pH units yr-1.**

As the estimated acidification rate is higher than the average ocean acidification it would be important to discuss the potential effects of the gaps in the times series some spanning also 7 consecutive years.

It would be relevant to compare the trends on periods without long gaps of data, which could strongly affect the slope of the trend.

**Following the reviewer's recommendation, the long-term trend in acidification was partially assessed in those periods best analysed, namely 1981-1998 and 2001-2009. The main conclusion of the results obtained was the loss of statistical significance of acidification rates. In fact, every interannual changes of deseasonalized pH time serie during the period 1981-1998 showed p-values> 0.05 while only the surface waters of the ocean and the continental shelf with pH trends of -0.0040±0.0006 yr$^{-1}$ and -0.0140±0.0017 yr$^{-1}$, respectively, were significant. In any case, we understand that the inclusion of the data shown in Table 2 beyond an intense study on the evolution of acidification in an upwelling system want to highlight the fact that direct observations in the Iberian Upwelling System over the last 40 years indicate that pH is decreasing.**

A comparison with other articles reporting ranges for coastal acidification trends could be interesting for improving the discussion of results.

**Other measures of acidification rates in near-shore areas such as the well-known ESTOC or CARIACO stations show year-on-year trends of -0.0018±0.0001 yr$^{-1}$ and -0.0025±0.0004 yr$^{-1}$, respectively. Other acidification rates like the one found at the DYFAMED station in the Mediterranean Sea was -0.0028±0.0003 yr$^{-1}$ (https://hal.sorbonne-universite.fr/hal-01534516/document). In general these rates of change are in the range of the information obtained from the ARIOS database as well as the acidification rates used as reference by Lauvset et al. 2015.**

**Coastal acidification includes local changes in water chemistry from changes in temperature or salinity, high nutrient inputs or inputs from freshwater rivers, or excess nutrient runoff (e.g., nitrogen and organic carbon). An ecosystem's ability to cope with acidification is influenced by the number of local stresses it faces. Some ecosystems may be more resilient to ocean acidification by minimizing biogeochemical changes. Because of these characteristics, the impact of acidification between coastal areas is difficult to compare, and extensive work is needed to analyse the similarities and differences between coastal areas.**

In the section "Cruises in the 2000s and recent years" (L. 261- 265), the information about each cruise is given but not always the months and years are given. I strongly suggest to provide similar information for each cruise or to refer to a more specific table where the time span of each cruise is given. In particular, regarding the last ARIOS project it is not clear in which months was carried out. **The number of days between the start and the end of the sampling period of each project was included in the Table 1.**

It would be important if the authors could be the precision for the temperature and salinity measurements in the period 1976-1984. **The information has been included in the new version of the manuscript. Namely, the precision of the temperature and salinity measurements in that period was 0.02ºC and 0.005 respectively.**

For chlorophyll measurements, as different filters were used, could the authors provide an estimation of the pore size given and of potential effects of the change. The indication of the volume filtered (range) could be also important if available. **The 6 cm Schleicher and Scholl and Whatman GF/F 2.5 cm filters are made of glass fibre and have a similar nominal pore of 0.7 micron. The Schleicher and Scholl filters were used for chlorophyll measurements with spectrophotometers that needed a larger sample volume because of their lower detection sensitivity while the Whatman filters always for fluoremetric measurements.**

Regarding the adopted Quality control procedure (L.370-L.382) it would be useful if the authors could provide a synthetic information on the first and second level of the quality control cited in this section.

Regarding the presented ARIOS data set there are some corrections to the data that the authors should consider as there are many negative concentrations for nitrites (n=4), nitrates (n=16), ammonia (n=13), and chlorophyll a (n=2). **The values of any parameter lower than the precision of those measurements as well as the negative measurements were replaced by zero and their corresponding flag by 6.**

There are concentrations for nutrients and chlorophyll a in the range of 10-3 to <10-7 that should be correctly reported, presumably, as less than the detection limits given in the methods, and properly flagged with QF = 6. **The values of any parameter lower than the precision of those measurements as well as the negative measurements were replaced by zero and their corresponding flag by 6.**

For all nutrients there are many values equal to 0 with QF= 2, these values presumably are below the detection limits and should be flagged with QF=6. **The values of any parameter lower than the precision of those measurements as well as the negative measurements were replaced by zero and their corresponding flag by 6.**

There are three in situ pH values in the range 7-7.6 that should be checked to evaluate if they can be considered reliable or doubtful. **These 3 measurements of pH were flagged = 3**

Below some minor comments are given:

L.52 I suggest to correct as follows: to help a sustainable management of the : : :
**The suggestion has been included in the new version of the manuscript**

L. 290 I think that "as well" should be omitted and I suggest to substitute "to create" with "to record".
**The suggestion has been included in the new version of the manuscript**

L. 306 "pH value to do so": unclear.
**The sentence was rewritten as follows: "the seawater pH measurements were determined with a spectrophotometric method following Clayton and Byrne (1993), subsequently adding 0.0047 to the pH value according to DelValls and Dickson (1998)."**

L325-327 it is unclear if for the titration the HCl concentration was 0.1 or 0.13 M.
**0.1M**

L.338, L.341 change the conjugation of verb to the past.
**The suggestion has been included in the new version of the manuscript**

L.351 "Cl2Mn" should be written as "MnCl2".
**The suggestion has been included in the new version of the manuscript**

L.352 change the conjugation of verb to the past.
**The suggestion has been included in the new version of the manuscript**

L. 382 check the year in the reference list is 2010.
**The correct year is 2010. The manuscript has been corrected.**

L. 506 "large seasonable variability" change with " large seasonal variability".
**The suggestion has been included in the new version of the manuscript**

L. 523 "and so it would" I suggest changing as: "and therefore it would".
**The suggestion has been included in the new version of the manuscript**

I suggest changing the yellow colour in Figure 1, as on the printed version is not clearly visible.
**The suggestion has been included in the new version of the manuscript. The new line is a green line.**

To enhance the readability of Figure 2, I suggest to enlarge them or to split the figure in two.

In Figures 3 and 4, for Salinity, I suggest omitting "psu" as it is not a real measurement unit, but a conductivity ratio. I suggest to indicate the pH is on the total scale similarly to figure 5.
**The pH on the total scale was indicated as $pH_T$ in every figure. The psu is mentioned to indicate salinity is reported in practical salinity scale. The new Figures 3 and 4 are attached to the reply to the referee.**

There is the need to correct the units of oxygen in micromole kg-1 in figure 4.
**The units of oxygen has been corrected.**

The subscript of pHT in the Figure 5 is not well readable and should be explained in the caption.

I suggest, all over the figures, to indicate the pH as pHT for clarifying that is expressed on the total scale, moreover, to increase the readability, I suggest to enlarge or split the two graphs.

TABLE 1. According to the data set, the ARIOS cruises were carried out during different months of 2018 and not in one month of 2017. As some cruises/projects could span over more month perhaps it would be better to provide the period of the study instead of a single date.
**The number of days (#d) between the start and the end of sampling period of each project was included in the Table 1 that is attached at the end of the reply to the referee.**

[revised manuscript text omitted]

---

## Author Response (AR2)

- Line 32: It is not good to start a sentence with a symbol. I suggest to write: Emission of anthropogenic origin CO2 ...

**The suggestion has been included in the new version of the manuscript**

- Line 49:IOC is a UNESCO body. To be more precise substitute United Nations with UNESCO

**The suggestion has been included in the new version of the manuscript**

- Line 326: following the suggestion of a referee, here I am recommending to change the sentence in: ... error was 0.003, using the equation of practical salinity given by UNESCO (1981) ...

**The suggestion has been included in the new version of the manuscript**

- Line 413: A quality flag ... Please, specify in parenthesis what QC flags convention are you using (SeaDataNet? WOCE? ...)

**The quality control flags in the previous version was a mixture of WOCE bottle flagging and SEADATANET codes, since following a reviewer's recommendation we assigned a flag of "6" to values below the detection limit. As we are using WHP-Exchange bottle format for the database, we believe it would be best to use an unique convection, namely WOCE bottle. Therefore, we will assign a code of "2" to those values that were flagged as "6" according to the SEADATANET flags. We will make this change in the repository, as soon as we know your opinion on this issue. The new sentence is:**

[revised manuscript text omitted]